# Right-to-Left Shunt Evaluation in Cardiac Patent Foramen Ovale Using Bubble Contrast Transcranial Color-Coded Doppler: A Cryptogenic Stroke Case

**DOI:** 10.3390/healthcare11192655

**Published:** 2023-09-29

**Authors:** Myeong-Hoon Ji, Youl-Hun Seoung

**Affiliations:** 1Department of Radiological Science, College of Health and Medical Sciences, Cheongju University, Cheongju 28503, Republic of Korea; bangour@cju.ac.kr; 2The Korean Registry for Diagnostic Medical Sonography (KRDMS), Daejeon 35041, Republic of Korea

**Keywords:** heart, patent foramen ovale, right-to-left shunt, bubble contrast, transcranial color-coded Doppler

## Abstract

Traditional diagnosis of patent foramen ovale (PFO) in the heart has involved the use of transcranial Doppler (TCD). However, TCD is essentially a blind test that cannot directly visualize the location of blood vessels. Since TCD relies on qualitative assessments by examiners, there is room for errors, such as misalignment of the ultrasound’s angle of incidence with the actual blood vessels. This limitation affects the reproducibility and consistency of the examination. In this study, we presented an alternative approach for assessing right-to-left shunt (RLS) associated with PFO using contrast transcranial color-coded Doppler (C-TCCD) with bubble contrast. The patient under consideration had been diagnosed with an ischemic stroke through imaging, but the subsequent cardiac work-up failed to determine the cause. Employing C-TCCD for RLS screening revealed a confirmed RLS of Spencer’s three grades. Subsequently, transesophageal echocardiography (TEE) was conducted to evaluate PFO risk factors, confirming an 8 mm PFO size, a 21 mm tunnel length, a hypermobile interatrial septum, and persistent RLS. The calculated high-risk PFO score was 4 points, categorizing it as a very high risk PFO. This case underscores the importance of C-TCCD screening in detecting RLS associated with PFO, especially in cryptogenic stroke patients, when identifying the underlying cause of ischemic stroke becomes challenging.

## 1. Introduction

Cryptogenic stroke (CS) refers to a situation in which an ischemic stroke occurs without a clearly identifiable cause, even after undergoing tests to search for an underlying trigger [1]. CS accounts for approximately 20% to 30% of all strokes, and its prevalence increases with age, affecting even younger individuals who are otherwise healthy [2]. Lately, a possible cause of CS has been linked to a right-to-left shunt (RLS) caused by a patent foramen ovale (PFO) in the heart. This occurs when there is heightened pressure in the right atrium compared to the left, such as during a cough [3]. This situation results in venous blood containing metabolites such as serotonin entering the arterial system and causing microemboli in the arterial system. This phenomenon can result in symptoms such as stroke or migraines [4]. In patients with CS, assessing RLS in PFOs is crucial in preventing secondary ischemic strokes [5]. Diagnosis involves inducing RLS movements through actions such as coughing or the Valsalva maneuver. These movements are difficult to immobilize for heart imaging techniques such as computed tomography (CT) or magnetic resonance imaging (MRI) [6].

The primary method for diagnosing RLS in PFOs relies on ultrasound examinations that provide real-time insights. Transesophageal echocardiography (TEE) is considered the gold standard for evaluating PFOs [7]. TEE facilitates the morphological assessment of PFO and has a high sensitivity of 89.2% and specificity of 91.4% in detecting RLS [8]. However, TEE necessitates the insertion of an endoscopic ultrasound probe into the patient’s esophagus, making it an invasive and uncomfortable procedure that may potentially require sedation [9]. Transcranial Doppler (TCD) is another method that uses a low-frequency probe to assess cerebral blood vessels through the temporal acoustic window. TCD-based RLS assessment involves injecting microbubbles of agitated saline contrast into venous blood and detecting high-intensity transient signals (HITS) in cerebral blood flow as the microbubbles travel into the arterial system [10,11]. TCD exhibits exceptional sensitivity (95%) and specificity (92%) in detecting RLS. It is simpler to perform than TEE and facilitates the execution of the Valsalva maneuver [12]. Additionally, studies indicate TEE has lower sensitivity and specificity than TCD when the PFO is small. Consequently, TEE has limitations in its utility screening for RLS in suspected CS patients. Instead, TCD is widely used in clinical settings [12,13,14,15,16,17]. Nonetheless, TCD is a blind examination method where vessel positioning is not directly observable. Its qualitative nature may lead to errors in aligning the patient’s actual vascular structure and ultrasound incidence angle, which can impact examination reproducibility [18,19].

The use of TCCD represents a significant advancement compared to conventional TCD. TCCD offers simultaneous B-scan and color Doppler imaging capabilities, enabling the direct visualization of vascular anatomical positions and providing real-time assessments of both morphological information and blood flow through color Doppler images. This advanced imaging technique has the potential to decrease errors related to subjective evaluations, thus enhancing the consistency and dependability of diagnoses for RLS. Furthermore, TCCD does not require additional equipment and can be conducted using existing ultrasound devices, making it highly accessible in clinical settings. This accessibility enhances its practicality for regular use in healthcare environments [20,21,22].

Thus, we attempted to utilize TCCD examination with bubble contrast in a real CS patient as a screening method for RLS. The aim of this case report is to provide clinicians with information on the application and usefulness of TCCD.

## 2. Case Report

### 2.1. Patient Characteristics

The case involves a 51-year-old man who presented to the neurosurgery outpatient department, complaining of weakness on the right side of his body one day before the consultation. His medical history included hypertension, but he was not regularly taking his medication. He had no other significant conditions, such as diabetes, tuberculosis, hepatitis, stroke, or cancer. There was no other significant family history. He had a 30-year history of smoking one pack of cigarettes daily and consuming one bottle of beer (500 mL) a day, three times a week. During the physical examination, he appeared alert and conscious, with a blood pressure of 180/116 mmHg, a heart rate of 118 beats per minute, a respiratory rate of 19 breaths per minute, and a body temperature of 36.0 degrees Celsius. His primary care physician detected no heart murmur during auscultation, while a neurological examination revealed decreased motor strength in his right arm and leg. Other organ systems appeared normal.

### 2.2. Diagnosis and Treatment of the Cause of CS in This Case Report

After brain imaging using a 3.0 Tesla magnetic resonance image machine (IGENIA CX 3.0T, Philips, Cambridge, MA, USA), the radiologist evaluated the CS and confirmed left acute hiatus infarction. The subsequent diagnostic process and treatment course for cerebral infarction are described in the following. Initially, the patient undergoes an echocardiogram and a 24-h electrocardiogram (ECG) recording to determine if the cause of the stroke is cardiac in origin. If the cause is non-cardiogenic, an evaluation of the RLS of the PFO is conducted. This procedure involves injecting a bubble contrast agent and performing a C-TCCD on the cranial temporal bone. If RLS is present, a TEE is performed to evaluate the PFO morphology and risk associated with it. A high-risk PFO indicates that closure will be recommended (Figure 1).

#### 2.2.1. Cardiac Testing

Echocardiography and 24-h ECG recordings were conducted to confirm the presence or absence of a cardiac source as the cause of the ischemic stroke. Echocardiography was performed using echocardiography (Affiniti 70C, Philips, USA) with a phased array probe (S5-1, Philips, USA) by a skilled registered diagnostic cardiac sonographer (RDCS). The echocardiogram revealed a normal heart chamber size, a heart wall thickness of 1.1 cm, normal cardiac function with a cardiac output of 56.3%, normal valves and morphology, and diastolic dysfunction (grade I). No pulmonary arterial hypertension, pericardial effusion, or intracardiac shunts were observed. A 24-h ECG (SEER 1000, General Electric, Boston, MA, USA) indicated normal heart rate with no bradycardia, supraventricular or ventricular beats, atrial fibrillation, or arrhythmia. Based on these results, the cardiologist diagnosed myocardial hypertrophy due to hypertension on cardiac examination and suspected RLS of the PFO as the cause of the stroke.

#### 2.2.2. Contrast-Enhanced TCCD

TCCD, employed with contrast enhancement, was utilized to evaluate the presence of RLS in PFO cases. The examination was conducted by a skilled international registration vascular technologist (RVT) who has more than ten years of practical experience in the field. To enhance the contrast in the images, a contrast agent was prepared using an agitated saline solution, commonly known as bubble saline.

The procedure for creating the contrast agent involved physically agitating 9 mL of saline solution along with 1 mL of air within a syringe. Swift mixing was achieved by utilizing a 3-way syringe stopcock (Figure 2).

This preparation method yielded micro-sized bubbles (Figure 3).

The effectiveness of the contrast agent was evaluated twice, once under resting conditions and once during stimulation induced using the Valsalva maneuver. For each test, the contrast agent was introduced into the median cubital vein.

This procedure was facilitated by imaging through the squamous section of the temporal bone. The detection of high-intensity transient signals (HITS), which indicate the presence of microbubbles, was conducted by assessing the middle segment of the right middle cerebral artery using TCCD (Figure 4).

The observation window for each scan was set to a duration exceeding 30 s, monitoring the occurrence of HITS within the vessel [24]. The assessment of RLS through the identification of HITS was quantified using the Spencer Grading Scale, a system that includes different grades. Each grade corresponds to a different degree of microbubble identification [25]. In this specific case, HITS were not detectable under resting conditions. However, during the Valsalva stimulation test, the presence of HITS was visually confirmed. A total of 37 instances of HITS were recorded, resulting in a grade 3 classification for the PFO-induced RLS. Consequently, the diagnosis of C-TCCD indicated the presence of an intracardiac short circuit caused by the PFO condition (Figure 5).

#### 2.2.3. Contrast-Enhanced Transesophageal Echocardiography

Contrast-enhanced TEE assessed the morphology of PFO and timing of RLS. TEE was employed with an ultrasound machine (Affiniti 70C, Philips, USA) and a probe (X7-2t, Philips, USA), which were operated by a cardiologist. A manufactured bubble contrast agent enhanced the images. TEE revealed a PFO diameter of approximately 8 mm, a tunnel length of approximately 21 mm, and a hypermobile interatrial septum (Figure 6).

Before the injection of contrast, no contrast was visible within the right atrium (Figure 7a). Following the administration of contrast, it became evident that the right atrium was filled with contrast (Figure 7b).

During the execution of the Valsalva maneuver, the contrast medium permeated into the left atrium (Figure 8). This occurrence confirmed the presence of hyperechoic microbubbles in both the left atrium and the RLS. Subsequently, an assessment of the high-risk PFO score yielded a value of 4, prompting the cardiologist to recommend PFO closure [26].

#### 2.2.4. Patent Foramen Ovale Closure

The PFO closure procedure was performed by a cardiologist. A five-French femoral sheath was introduced through the right femoral vein and subsequently replaced with a guide system (Amplatzer Trevisio 45 Delivery System, Amplater, Plymouth, MN, USA) designed to facilitate the transportation of the closure device to the heart. The selected closure device was an Amplatzer PFO occluder, which measured 25 mm. Utilizing TEE in conjunction with the guidance system, the device was accurately positioned at the heart’s PFO (Figure 9a). The closure of the PFO was then accomplished (Figure 9b). Following the confirmation of successful closure, the device was detached, marking the completion of the procedure. After the procedural steps, echocardiography and plain chest imaging were performed to confirm the accurate positioning and alignment of the closure device, as well as to identify any abnormalities in the surrounding structures. The procedure concluded without any observed irregularities. There were no procedural complications or recurrent strokes during the follow-up period.

## 3. Discussion

PFOs, affecting approximately 25% to 35% of the population [26], are one of the most common types of heart septal defects. Despite this, most individuals with PFOs remain asymptomatic and lead normal lives. Conversely, the incidence of PFO surpasses 40% [27] among stroke patients. When a PFO gives rise to RLS, it can lead to various diverse clinical complications, including migraine and stroke [4]. Although the recurrence rate of ischemic stroke attributed to RLS is below 2%, which is relatively low compared to other causes, its cumulative incidence is noteworthy due to its manifestation at an early age [28]. Notably, if the PFO measures over 2 mm or if RLS persists even during rest, the annual recurrence rate can escalate to 15% [29]. In this case, the patient had a long history of smoking and hypertension but did not exhibit significant clinical symptoms. The initial MRI scan unveiled an acute left lacunar cerebral infarction, which was ultimately diagnosed as an ischemic stroke. However, further neurological evaluations could not pinpoint the origin of the stroke. Consequently, a cardiology consultation was sought to rule out the presence of cardiogenic thrombus.

The echocardiography illustrated an increase in ventricular wall thickness to 1.1 cm, indicating cardiac remodeling due to prolonged exposure to high blood pressure [30,31]. Additionally, ventricular diastolic dysfunction was attributed to age-related changes [32]. Cardiogenic thrombi form as a result of reduced cardiac output, atrial fibrillation, and intracardiac thrombi, which occur due to the hemodynamic stagnation of blood within the heart [33]. Nonetheless, 24-h ECG monitoring detected no arrhythmia or atrial fibrillation, thus ruling out the possibility of universal cardiogenic thrombi involvement.

A suspicion of RLS due to PFO prompted the cardiologist to order further C-TCCD testing. Results displayed 37 HITS using the Valsalva method with RLS and a Spencer grade of 3. The Spencer scale, which measures RLS using cerebral blood flow ultrasound, employs graded levels determined by counting HITS that occur within 30 s of injecting the contrast medium. In this instance, the PFO was determined to be moderate in size, with a Spencer grade of 3 [25].

The evaluation of the PFO through TEE revealed a size of approximately 8 mm, with the tunnel spanning about 21 mm. Real-time imaging revealed the hyperkinetic cardiac septum and the flow of contrast agents through the PFO. This morphological assessment via TEE resulted in a high-risk PFO score evaluation, which serves as the basis for determining closure. A 2019 study involving 107 patients with PFO undergoing TEE identified five factors linked to an increased risk of CS. These factors were used to develop a scoring system for classifying high-risk PFO, assigning one point for each factor present. The factors include (1) PFO tunnel length exceeding 10 mm, (2) presence of a hyperkinetic heart septum, (3) presence of Eustachian valve or Chiari’s network, (4) substantial RLS during Valsalva maneuver, and (5) septum-to-PFO angle less than or equal to 10 degrees. The study showed that a score of 2 or more correlated with high sensitivity (91%) and specificity (80%) for the association between PFO and CS [26]. Thus, in this scenario, a high-risk PFO score of 4 indicates significant risk. Considering moderate RLS and a morphological assessment that resulted in a risk score of 4 or higher through TEE, the cardiologist decided to proceed with a successful PFO closure.

There are two primary options for preventing recurrent stroke in patients with CS and PFO: related RLS medication and PFO closure. A prospective randomized controlled study in 2012 assessing both treatments found no discernible difference [34]. However, subsequent studies have highlighted the advantages of closure in preventing recurrent stroke [35,36]. Nevertheless, we need a comprehensive assessment in the decision-making process of the potential downsides of PFO closure, including device-induced thrombosis, invasive procedures, and elevated atrial fibrillation rates [37]. In light of this, recently, an emphasis has been placed on screening for RLS in CS patients with PFOs.

While both domestic and international studies predominantly employ TCD for RLS evaluation, TCD has limitations, such as its inability to directly visualize vessels and its susceptibility to patient movement. Particularly, C-TCD-based RLS assessment has a 30-s time constraint post-contrast injection, which could potentially result in missed blood flow due to patient motion. In contrast, TCCD directly observes the locations of blood vessels, allowing for quick re-measurement even if blood flow is momentarily disrupted by movement. Consequently, TCCD, in contrast, holds promise for enhanced evaluation of PFO and RLS. Nevertheless, contrast-enhanced TCCD lacks standardization in Korea. Therefore, this case report confirms the usefulness of contrast-enhanced TCCD, and further clinical studies should be conducted.

Limitations of this case report included potential bias resulting from the testing and procedures conducted on a single patient. In addition, the short follow-up period of the case made it difficult to observe and evaluate the long-term progression of clinical symptoms.

## 4. Conclusions

In this case analysis, we suggested a report on a PFO occlusion following RLS screening of a CS patient using C-TCCD and TEE.

The TCCD method, which utilizes a bubble contrast agent for screening RLS in PFO, showed a faster ability to handle interruptions in blood flow signals caused by patient movement compared to the established TCD test method.The assessment of RLS using C-TCCD and TEE, along with the morphological examination of the PFO, played a crucial role in determining the need for PFO closure.The significance of C-TCCD screening in identifying RLS induced by PFO was reaffirmed in patients with CS who are struggling to determine the cause of their ischemic stroke.In this case, the patient was young and was found to have a PFO with characteristics that suggest a high risk. Consequently, the initial therapeutic approach prioritized closing the PFO over pharmacological intervention.

## Figures and Tables

**Figure 1 healthcare-11-02655-f001:**
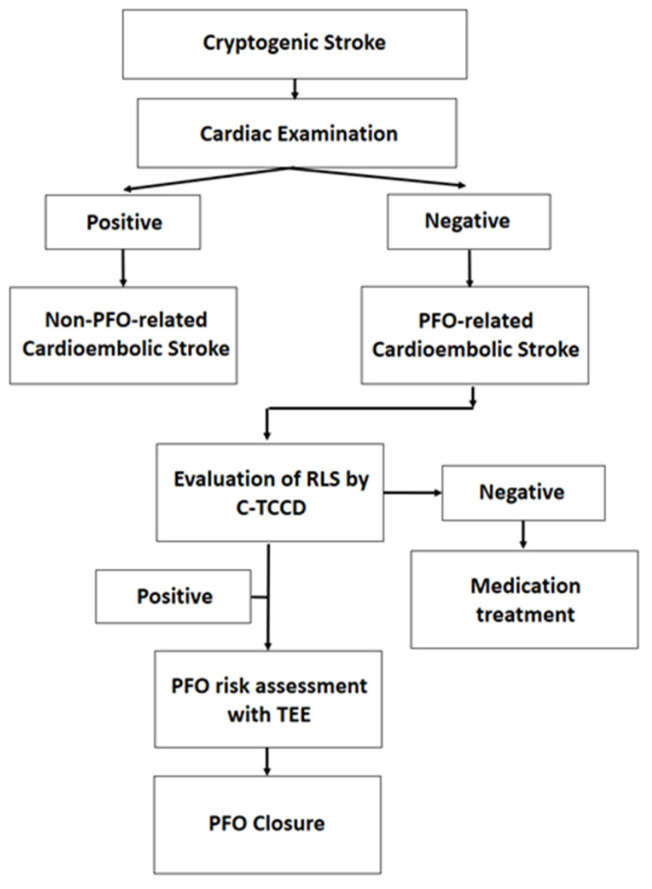
Assessment of PFO closure eligibility flowchart in this case report.

**Figure 2 healthcare-11-02655-f002:**
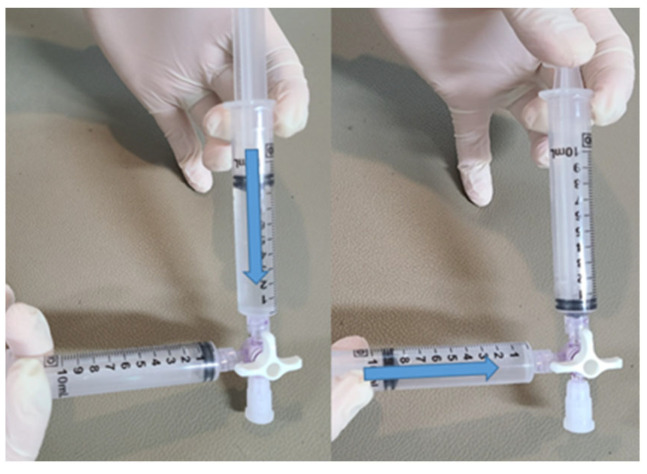
The process of making agitated saline via physical reciprocation.

**Figure 3 healthcare-11-02655-f003:**
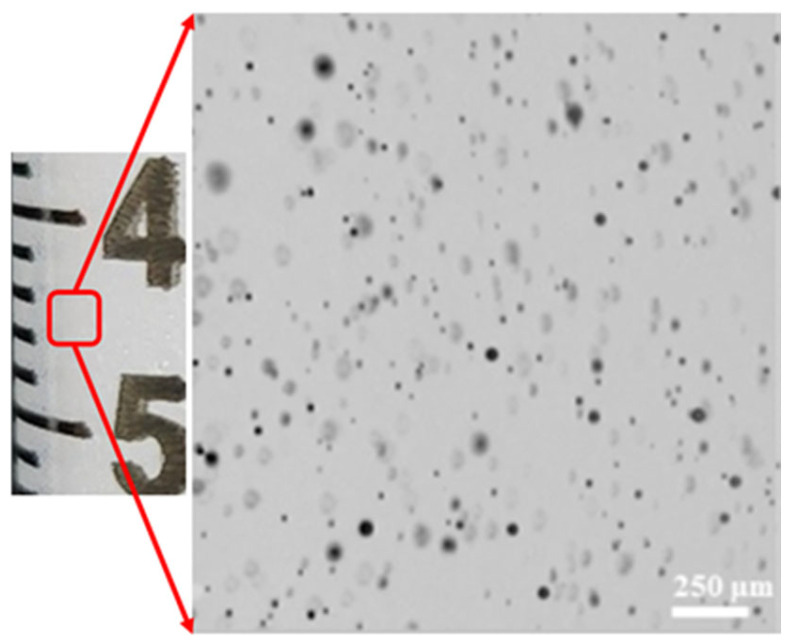
Microbubbles in agitated saline generated via a reciprocating method (microscopic image borrowed from Authorea [23]).

**Figure 4 healthcare-11-02655-f004:**
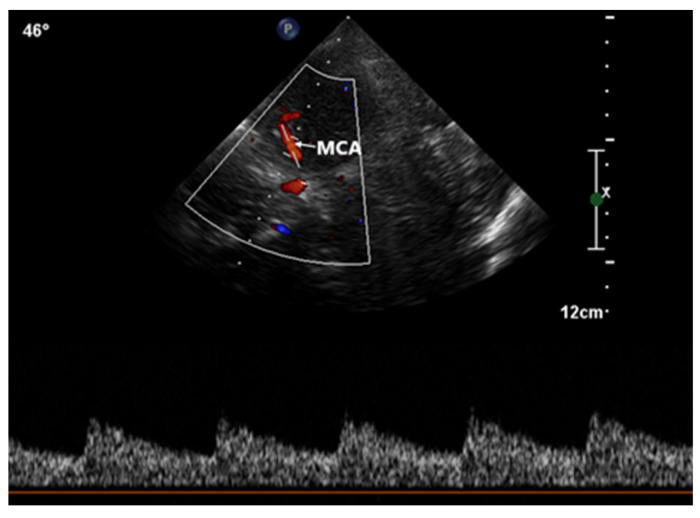
Transcranial color-coded Doppler with pulsed-wave spectral Doppler of the middle cerebral artery. Abbreviation: MCA, middle cerebral artery.

**Figure 5 healthcare-11-02655-f005:**
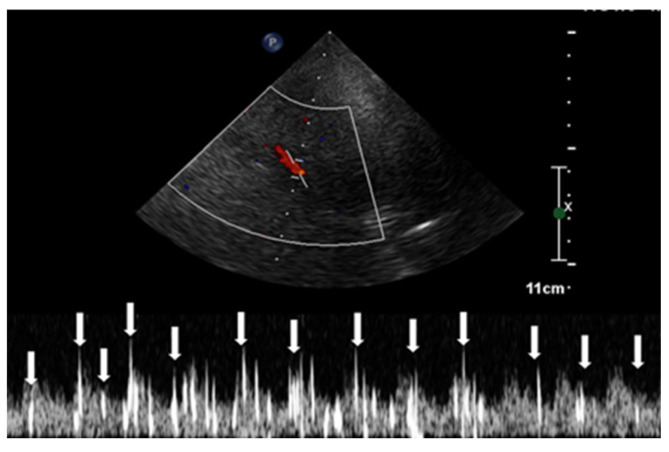
High-intensity transient signals (HITS) caused by microbubbles in the TCCD contrast and right-to-left shunt positivity.

**Figure 6 healthcare-11-02655-f006:**
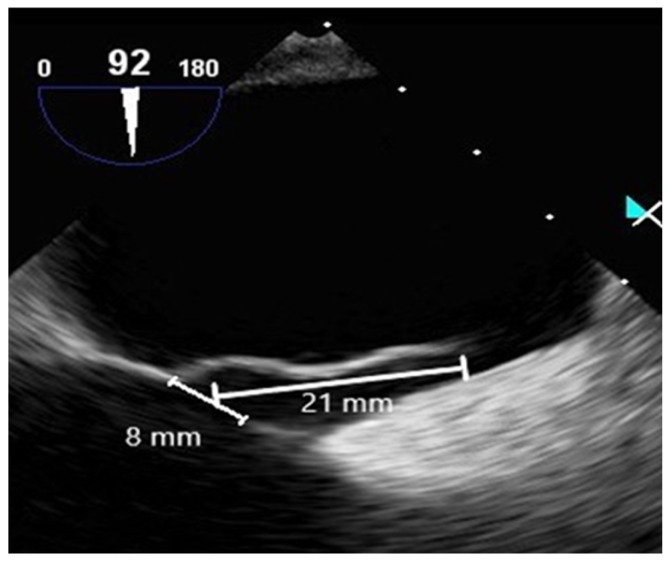
TEE view of measured PFO diameter (8 mm) and tunnel length (21 mm).

**Figure 7 healthcare-11-02655-f007:**
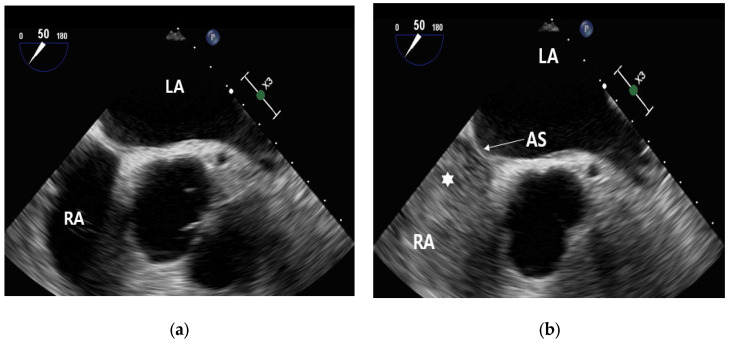
Contrast enhancement in the right atrium. Pre- and post-injection observations: (**a**) microbubble pre-contrast injection and (**b**) filled with microbubble contrast (★) in the right atrium. Abbreviation: LA, left atrium; RA, Right atrium; AS, atrial septum.

**Figure 8 healthcare-11-02655-f008:**
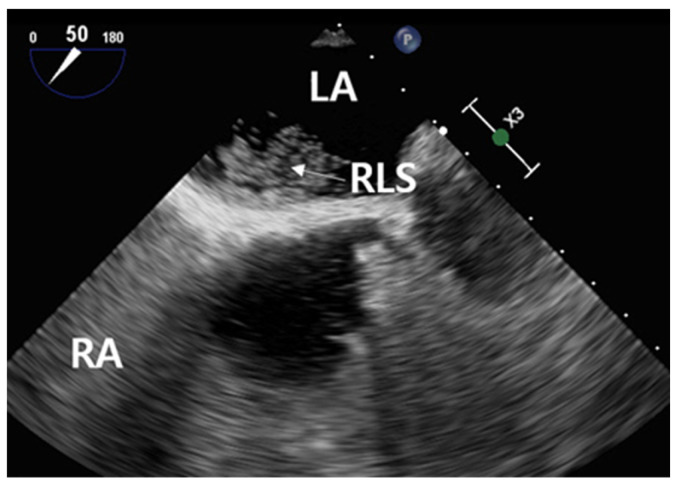
TEE image of RLS following stimulation using the Valsalva maneuver. Abbreviation: LA, left atrium; RA, right atrium; RLS, right to left shunt.

**Figure 9 healthcare-11-02655-f009:**
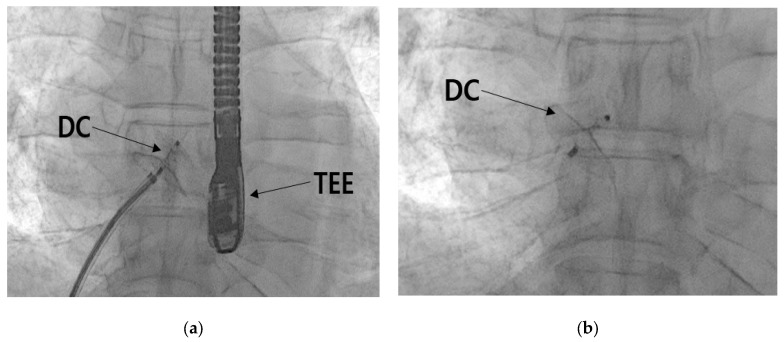
Process of intervention to close the PFO. (**a**) Position of the PFO device closure while guided by the transesophageal echocardiography, and (**b**) the PFO device closure removed from the guide system. Abbreviation: DC, device closure; TEE, transesophaheal echocardiography.

## Data Availability

Not applicable.

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
