# Peer review of "Right-to-Left Shunt Evaluation in Cardiac Patent Foramen Ovale Using Bubble Contrast Transcranial Color-Coded Doppler: A Cryptogenic Stroke Case"

_healthcare, 2023, doi:10.3390/healthcare11192655_

Round 1
Reviewer 1 Report
I reviewed with interest the manuscript "Right to Left Shunt Evaluation in Cardiac Patent Foramen Ovale Using Bubble Contrast Transcranial Color Coded Doppler: Focused on Cryptogenic Stroke Case" by Myeong-Hoon Ji and Youl-Hun Seoung. In this article, the authors present a clinical case of successful diagnosis of Patent Foramen Ovale Using Bubble Contrast Transcranial Color Coded Doppler. Although the technique of using Transcranial Doppler with bubble contrast to diagnose patent foramen ovale is well known (1), the authors used Transcranial Color Coded Doppler for the same purpose (apparently, one of the first). This makes it possible to improve the diagnostic process due to better positioning of the transcranial sensor along the vessel. This result may be of practical use in clinical settings. However, during the review I had questions and comments to which I would like to receive answers from the authors in this section.
1. The list of references contains many references to old publications (11 are older than 5 years, 13 are older than 13 years). It is difficult to explain the relevance of this publication with such old links. I think that more recent publications on this topic should be cited (for example, links 2-4, see below)
2. Authors' statement in the Introduction section "A PFO is a hole in the septum, which is a thin muscular wall that separates the right and left atrium. RLS refers to a momentary shift of blood flow from the right to the left side." (lines 33-35) states a well-known fact and is unnecessary for an article in a scientific journal.
3. The authors’ phrase “This situation prompts venous blood to flow into the arterial system, leading to microembolisms within the venous system...” (lines 36-38) is inaccurate and requires correction.
4. The authors’ phrase “... consuming one bottle of alcohol three times a week” is non-particular. If the authors decided to give an alcohol history, then it requires specification (the bottle could be beer, wine, whiskey, etc.). Can you provide information in drinks?
5. In section "2. CS cause diagnosis and treatment process" the authors describe the diagnostic and treatment algorithm for CS. I don't think this is appropriate when describing a clinical case. It would be more appropriate to place this section at the end of the Discussion section as a suggestion from the authors of the article for the future.
6. The phrase “The cardiologist diagnosed the stroke as myocardial hypertrophy due to hypertension and attributed the RLS of the PFO as the cause of the stroke” (lines 1207-109) is unclear and requires correction.
7. There is no reference in the text to source 23, to Figure 8.
8. The authors write “Thus, this case underscores the necessity for standardized clinical protocols for contrast-enhanced TCCD in RLS assessment...” (lines 241-243). In my opinion, based on only one clinical case, it is somewhat early to propose a standardized clinical protocol; further research is still necessary first.
References:
1. Goutman SA, Katzan IL, Gupta R. Transcranial Doppler with bubble study as a method to detect extracardiac right-to-left shunts in patients with ischemic stroke. J Neuroimaging. 2013 Oct;23(4):523-5. doi: 10.1111/j.1552-6569.2012.00738.x.
2. Zetola VF, Lange MC, Scavasine VC, et al. Latin American Consensus Statement for the Use of Contrast-Enhanced Transcranial Ultrasound as a Diagnostic Test for Detection of Right-to-Left Shunt. Cerebrovasc Dis. 2019;48(3-6):99-108. doi: 10.1159/000503851.
3. Khurana D, Petluri G, Kumar M, et al. Prevalence of Patent Foramen Ovale in North Indian Cryptogenic Young Strokes. Neurol India. 2022 May-Jun;70(3):1077-1082. doi: 10.4103/0028-3886.349647.
4. Shi F, Sha L, Li H, et al. Recent progress in patent foramen ovale and related neurological diseases: A narrative review. Front Neurol. 2023 Mar 27;14:1129062. doi: 10.3389/fneur.2023.1129062.
No comments
Reviewer 2 Report
While the case report provides valuable information about the patient's medical history, diagnosis, and treatment process, there are several negative aspects that should be addressed:
1) The authors should clearly state the research objectives or the hypothesis being tested in the introduction section.
2) Also, the novelty of study, why it is important, and potential impact on clinical practice should be mentioned in the introduction section, since the article is a case-report.
3) While the report mentions the patient's hypertension, it lacks a comprehensive overview of the patient's medical history, including any medications he may have been taking or any family history of stroke or other relevant conditions. A more thorough patient history would provide a more comprehensive understanding of the case.
4) The report does not sufficiently justify the choice of treatment (PFO closure) based on the diagnostic findings (in the results section). It should include a clear rationale for why this specific treatment was chosen, at least briefly in the results section.
5) The report does not provide information about the patient's clinical outcomes following the PFO closure. It should include details about the patient's recovery, any complications, and long-term follow-up.
6) A discussion of the risks and benefits associated with the PFO closure procedure should be included. This would help readers understand the potential advantages and drawbacks of the treatment.
7) The report should acknowledge any limitations of the case, such as potential biases or constraints in data collection and analysis. Addressing these limitations would enhance the transparency of the study.
Round 2
Reviewer 1 Report
The authors answered my questions and comments and made corrections to the text of the manuscript. I have no other comments.
No comments
Reviewer 2 Report
The authors revised the manuscript taking into account the suggestions we made and thus making a better understanding of the presented case.